# EqCo: Equivalent Rules for Self-supervised Contrastive Learning

## Abstract

In this paper, we propose a method, named EqCo (**Eq**uivalent Rules for **Co**ntrastive Learning), to make self-supervised learning irrelevant to the number of negative samples in the contrastive learning framework. Inspired by the InfoMax principle, we point that the margin term in contrastive loss needs to be adaptively scaled according to the number of negative pairs in order to keep steady mutual information bound and gradient magnitude. EqCo bridges the performance gap among a wide range of negative sample sizes, so that we can use only a few negative pairs (e.g. 16 per query) to perform self-supervised contrastive training on large-scale vision datasets like ImageNet, while with almost no accuracy drop. This is quite a contrast to the widely used large batch training or memory bank mechanism in current practices. Equipped with EqCo, our simplified MoCo (SiMo) achieves comparable accuracy with MoCo v2 on ImageNet (linear evaluation protocol) while only involves 16 negative pairs per query instead of 65536, suggesting that large quantities of negative samples might not be a critical factor in contrastive learning frameworks.

## 1 Introduction and Background

Self-supervised learning has recently received much attention in the field of visual representation learning (Hadsell et al. (2006); Dosovitskiy et al. (2014); Oord et al. (2018); Bachman et al. (2019); Hénaff et al. (2019); Wu et al. (2018); Tian et al. (2019); He et al. (2020); Misra & Maaten (2020); Grill et al. (2020); Cao et al. (2020); Tian et al. (2020)), as its potential to learn universal representations from unlabeled data. Among various self-supervised methods, one of the most promising research paths is *contrastive learning* (Oord et al. (2018)), which has been demonstrated to achieve comparable or even better performances than supervised training for many downstream tasks such as image classification, object detection, and semantic segmentation (Chen et al., 2020c; He et al., 2020; Chen et al., 2020a;b).

The core idea of contrastive learning is briefly summarized as follows: first, extracting a pair of embedding vectors $(\mathbf{q}(I), \mathbf{k}(I))$ (named *query* and *key* respectively) from the two augmented views of each instance $I$; then, learning to maximize the similarity of each *positive pair* $(\mathbf{q}(I), \mathbf{k}(I))$ while pushing the *negative pairs* $(\mathbf{q}(I), \mathbf{k}(I'))$ (i.e., query and key extracted from different instances accordingly) away from each other. To learn the representation, an *InfoNCE loss* (Oord et al. (2018); Wu et al. (2018)) is conventionally employed in the following formulation (slightly modified with an additional *margin* term):

$$\mathcal{L}_{NCE} = \mathop{\mathbb{E}}_{\mathbf{q}\sim\mathcal{D},\mathbf{k}_0\sim\mathcal{D}'(\mathbf{q}),\mathbf{k}_i\sim\mathcal{D}'} \left[ -\log\frac{e^{(\mathbf{q}^\top\mathbf{k}_0-m)/\tau}}{e^{(\mathbf{q}^\top\mathbf{k}_0-m)/\tau} + \sum_{i=1}^{K}e^{\mathbf{q}^\top\mathbf{k}_i/\tau}} \right], \tag{1}$$

where $\mathbf{q}$ and $\mathbf{k}_i$ ($i = 0, \ldots, K$) stand for the query and keys sampled from the two (augmented) data distributions $\mathcal{D}$ and $\mathcal{D}'$ respectively. Specifically, $\mathbf{k}_0$ is associated to the same instance as $\mathbf{q}$'s while other $\mathbf{k}_i$s not; hence we name $\mathbf{k}_0$ and $\mathbf{k}_i$ ($i > 0$) *positive sample* and *negative samples* respectively in the remaining text, in which $K$ is the number of negative samples (or pairs) for each query. The *temperature* $\tau$ and the *margin* $m$ are hyper-parameters. In most previous works, $m$ is trivially set to zero (e.g. Oord et al. (2018); He et al. (2020); Chen et al. (2020a); Tian et al. (2020)) or some

handcraft values (e.g. Xie et al. (2020)). In the following text, we mainly study contrastive learning frameworks with InfoNCE loss as in Eq. 1 unless otherwise specified. [1]

In contrastive learning research, it has been widely believed that enlarging the number of negative samples $K$ boosts the performance (Hénaff et al. (2019); Tian et al. (2019); Bachman et al. (2019)). For example, in MoCo (He et al. (2020)) the ImageNet accuracy rises from 54.7% to 60.6% under linear classification protocol when $K$ grows from 256 to 65536. Such observation further drives a line of studies how to effectively optimize under a number of negative pairs, such as *memory bank* methods (Wu et al. (2018); He et al. (2020)) and large batch training (Chen et al. (2020a)), either of which empirically reports superior performances when $K$ becomes large. Analogously, in the field of *supervised metric learning* (Deng et al. (2019); Wang et al. (2018); Sun et al. (2020); Wang et al. (2020)), loss in the similar form as Eq. 1 is often applied on a lot of negative pairs for hard negative mining. Besides, there are also a few theoretical studies supporting the viewpoint. For instance, Oord et al. (2018) points out that the mutual information between the positive pair tends to increase with the number of negative pairs $K$; Wang & Isola (2020) find that the negative pairs encourage features' uniformity on the hypersphere; Chuang et al. (2020) suggests that large $K$ leads to more precise estimation of the debiased contrastive loss; etc.

Despite the above empirical or theoretical evidence, however, we point out that the reason for using many negative pairs is still less convincing. **First**, unlike the metric learning mentioned above, in self-supervised learning, the negative terms $\mathbf{k}_i$ in Eq. 1 include both "true negative" (whose underlying class label is different from the query's, similarly hereinafter) and "false negative" samples, since the actual ground truth label is not available. So, intuitively large K should not always be beneficial because the risk of false negative samples also increases (known as *class collision* problem). Arora et al. (2019) thus theoretically concludes that a large number of negative samples could not necessarily help. **Second**, some recent works have proven that by introducing new architectures (*e.g.*, a predictor network in BYOL (Grill et al., 2020)), or designing new loss functions (*e.g.*, Caron et al. (2020a); Ermolov et al. (2020)), state-of-the-art performance can still be obtained even without any explicit negative pairs. In conclusion, it is still an open question whether large quantities of negative samples are essential to contrastive learning.

After referring to the above two aspects, we rise a question: **is a large $K$ really essential in the contrastive learning framework**? We propose to rethink the question from a different view: note that in Eq. 1, there are three hyper-parameters: the number of negative samples $K$, temperature $\tau$, and margin $m$. In most of previous empirical studies (He et al. (2020); Chen et al. (2020a)), only $K$ is changed while $\tau$ and $m$ are usually kept constant. *Do the optimal hyper-parameters of $\tau$ and $m$ varies with $K$?* If so, the performance gains observed from larger $K$s may be a *wrong* interpretation – merely brought by suboptimal hyper-parameters' choices for small $K$s, rather than much of an essential.

In the paper, we investigate the relationship among three hyper-parameters and suggest an *equivalent rule*:

$$m = \tau \log \frac{\alpha}{K},$$

where $\alpha$ is a constant. We find that if the margin $m$ is adaptively adjusted based on the above rule, the performance of contrastive learning is *irrelevant* to the size of $K$, in a very large range (e.g. $K \geq 16$). For example, in MoCo framework, by introducing EqCo the performance gap between $K = 256$ and $K = 65536$ (the best configuration reported in He et al. (2020)) *almost disappears* (from 6.1% decrease to 0.2%). We call this method "**Eq**uivalent Rules for **Co**ntrastive learning" (*EqCo*). For completeness, as the other part of EqCo we point that adjusting the learning rate according to the conventional *linear scaling rule* satisfies the equivalence for different number of *queries* per batch.

Theoretically, following the *InfoMax principle* (Linsker (1988)) and the derivation in *CPC* (Oord et al. (2018)), we prove that in *EqCo*, the lower bound of the mutual information keeps steady under various numbers of negative samples $K$. Moreover, from the back-propagation perspective, we further prove that in such configuration the upper bound of the gradient norm is also free of

---

[1]Recently, some self-supervised learning algorithms achieve new state-of-the-art results using different frameworks instead of conventional *InfoNCE* loss as in Eq. 1, e.g. *mean teacher* (in *BYOL* Grill et al. (2020)) and *online clustering* (in *SWAV* Caron et al. (2020b)). We will investigate them in the future.

$K$'s scale. The proposed equivalent rule implies that, by assigning $\alpha = K_0$, it can "mimic" the optimization behavior under $K_0$ negative samples even if the *physical* number of negatives $K \neq K_0$.

The "equivalent" methodology of EqCo follows the well-known *linear scaling rule* (Krizhevsky (2014); Goyal et al. (2017)), which suggests scaling the learning rate proportional to the batch size if the loss satisfies with the linear averaged form: $L = \frac{1}{N}\sum_{i=1}^{N} f(x_i; \theta)$. However, *linear scaling rule* cannot be directly applied on InfoNCE loss (Eq. 1), which is partially because InfoNCE loss includes two batch sizes (number of *queries* and *keys* respectively) while linear scaling rule only involves one, in addition to the nonlinearity of the keys in InfoNCE loss. In the experiments of *SimCLR* (Chen et al. (2020a)), learning rates under different batch sizes are adjusted with linear scaling rule, but the accuracy gap is still very large (57.5%@batch=256 vs. 64+%@batch=8192, 100 epochs training).

EqCo challenges the belief that self-supervised contrastive learning requires large quantities of negative pairs to obtain competitive performance, making it possible to design simpler algorithms. We thus present *SiMo*, a simplified contrastive learning framework based on *MoCo v2* (Chen et al. (2020c)). SiMo is elegant, efficient, free of large batch training and memory bank; moreover, it can achieve superior performances over state-of-the-art even if the number of negative pairs is extremely small (e.g. 16), without bells and whistles.

The contributions of our paper are summarized as follows:

- We challenge the widely accepted belief that on large-scale vision datasets like ImageNet, large size of negative samples is critical for contrastive learning. We interpret it from a different view: it may be because the hyper-parameters are not set to the optimum.

- We propose EqCo, an equivalent rule to adaptively set hyper-parameters between small and large numbers of negative samples, which proves to bridge the performance gap.

- We present SiMo, a simpler but stronger baseline for contrastive learning.

## 2 EqCo: Equivalent Rules for Contrastive Learning

In this section we introduce *EqCo*. We mainly consider the circumstance of optimizing the *InfoNCE loss* (Eq. 1) with *SGD*. For each batch of training, there are two meanings of the concept "batch size", i.e., the size of *negative samples/pairs* $K$ per query, and the number of *queries* (or *positive pairs*) $N$ per batch. Hence our equivalent rules accordingly consist of two parts, which will be introduced in the next subsections.

### 2.1 The Case of Negative Pairs

Our derivation is mainly inspired by the model of *Contrastive Predictive Coding (CPC)* (Oord et al. (2018)), in which *InfoNCE loss* is interpreted as a mutual information estimator. We further extend the method so that it is applicable to InfoNCE loss with a *margin* term (Eq. 1), which is not considered in Oord et al. (2018).

Following the concept in Oord et al. (2018), given a *query* embedding $\mathbf{q}$ (namely the *context* in Oord et al. (2018)) and suppose $K + 1$ random *key* embeddings $\mathbf{x} = \{\mathbf{x}_i\}_{i=0,\ldots,K}$, where there exists exactly one entry (e.g., $\mathbf{x}_i$) sampled from the conditional distribution $P(\mathbf{x}_i|\mathbf{q})$ while others (e.g., $\mathbf{x}_j$) sampled from the "proposal" distribution $P(\mathbf{x}_j)$ independently. According to which entry corresponds to the conditional distribution, we therefore defines $K + 1$ *candidate* distributions for $\mathbf{x}$ (denoted by $\{H_i\}_{i=0,\ldots,K}$), where the probability density of $\mathbf{x}$ under $H_i$ is $P_{H_i}(\mathbf{x}) = P(\mathbf{x}_i|\mathbf{q})\prod_{j\neq i} P(\mathbf{x}_j)$. So, given the observed data $X = \{\mathbf{k}_0,\ldots,\mathbf{k}_K\}$ of $\mathbf{x}$, the probability where $\mathbf{x}$ is sampled from $H_0$ rather than other candidates is thus derived with Bayes theorem:

$$
\begin{aligned}
\Pr[\mathbf{x} \sim H_0 | \mathbf{q}, X] &= \frac{P^+ P_{H_0}(X)}{P^+ P_{H_0}(X) + P^- \sum_{i=1}^{K} P_{H_i}(X)} \\
&= \frac{\frac{P^+}{P^-}\frac{P(\mathbf{k}_0|\mathbf{q})}{P(\mathbf{k}_0)}}{\frac{P^+}{P^-}\frac{P(\mathbf{k}_0|\mathbf{q})}{P(\mathbf{k}_0)} + \sum_{i=1}^{K}\frac{P(\mathbf{k}_i|\mathbf{q})}{P(\mathbf{k}_i)}},
\end{aligned}
\tag{2}
$$

where we denote $P^+$ and $P^-$ as the *prior* probabilities of $H_0$ and $H_i(i > 0)$ respectively. We point that Eq. 2 introduces a *generalized form* to that in Oord et al. (2018) by taking the priors into account. Referring to the notations in Eq. 1, we suppose that $H_0$ is the ground truth distribution of $\mathbf{x}$ (since $\mathbf{k}_0$ is the only positive sample). By modeling the density ratio $P(\mathbf{k}_i|\mathbf{q})/P(\mathbf{k}_i) \propto e^{\mathbf{q}^\top \mathbf{k}_i/\tau}(i = 0, \ldots, K)$ and letting $P^+/P^- = e^{-m/\tau}$, the negative log-likelihood $\mathcal{L}_{opt} \triangleq \mathbb{E}_{\mathbf{q},X} - \log \Pr[x \sim H_0|\mathbf{q}, X]$ can be regarded as the optimal value of $\mathcal{L}_{NCE}$.

Similar to the methodology of Oord et al. (2018), we explore the lower bound of $\mathcal{L}_{opt}$:

$$
\begin{aligned}
\mathcal{L}_{opt} &= \mathop{\mathbb{E}}_{\mathbf{q}\sim\mathcal{D},\mathbf{k}_0\sim\mathcal{D}'(\mathbf{q}),\mathbf{k}_i\sim\mathcal{D}'} \log\left(1 + e^{m/\tau}\frac{P(\mathbf{k}_0)}{P(\mathbf{k}_0|\mathbf{q})}\sum_{i=1}^{K}\frac{P(\mathbf{k}_i|\mathbf{q})}{P(\mathbf{k}_i)}\right) \\
&\approx \mathop{\mathbb{E}}_{\mathbf{q}\sim\mathcal{D},\mathbf{k}_0\sim\mathcal{D}'(\mathbf{q})} \log\left(1 + Ke^{m/\tau}\frac{P(\mathbf{k}_0)}{P(\mathbf{k}_0|\mathbf{q})}\left(\mathop{\mathbb{E}}_{\mathbf{k}_i\sim\mathcal{D}'}\frac{P(\mathbf{k}_i|\mathbf{q})}{P(\mathbf{k}_i)}\right)\right) \\
&= \mathop{\mathbb{E}}_{\mathbf{q}\sim\mathcal{D},\mathbf{k}_0\sim\mathcal{D}'(\mathbf{q})} \log\left(1 + Ke^{m/\tau}\frac{P(\mathbf{k}_0)}{P(\mathbf{k}_0|\mathbf{q})}\right) \\
&\geq \log(1 + Ke^{m/\tau}) - \mathcal{I}(\mathbf{k}_0, \mathbf{q}),
\end{aligned}
\tag{3}
$$

where $\mathcal{I}(\cdot, \cdot)$ means mutual information. The approximation in the second row is guaranteed by *Law of Large Numbers* as well as the fact $P(\mathbf{k}_i|\mathbf{q}) \approx P(\mathbf{k}_i)$ since $\mathbf{k}_i(i > 0)$ and $\mathbf{q}$ are "almost" independent. The inequality in the last row is resulted from $P(\mathbf{k}_0|\mathbf{q}) \geq P(\mathbf{k}_0)$ as $\mathbf{k}_0$ and $\mathbf{q}$ are extracted from the same instance. Therefore the lower bound of the mutual information (noted as $f_{\text{bound}}(m, K)$) between the *positive pair* $(\mathbf{k}_0, \mathbf{q})$ is:

$$
\begin{aligned}
\mathcal{I}(\mathbf{k}_0, \mathbf{q}) \geq f_{\text{bound}}(m, K) &\triangleq \log(1 + Ke^{m/\tau}) - \mathcal{L}_{opt} \\
&\approx \log(1 + Ke^{m/\tau}) - \mathop{\mathbb{E}}_{\mathbf{q}\sim\mathcal{D},\mathbf{k}_0\sim\mathcal{D}'(\mathbf{q})} \log\left(1 + Ke^{m/\tau}\frac{P(\mathbf{k}_0)}{P(\mathbf{k}_0|\mathbf{q})}\right).
\end{aligned}
\tag{4}
$$

So, minimizing $\mathcal{L}_{NCE}$ (Eq. 1) towards $\mathcal{L}_{opt}$ implies maximizing the lower bound of the mutual information, **which is also satisfied when $m \neq 0$**. In the case of $m = 0$, the result is consistent with that in Oord et al. (2018). Oord et al. (2018) further points out the bound increases with $K$, which indicates larger $K$ encourages to learn more mutual information thus could help to improve the performance.

Nevertheless, different from Oord et al. (2018) our model does not require $m$ to be zero, so the lower bound in Eq. 4 is also a function of $e^{m/\tau}$. Thus we have the following theorem:

**Theorem 1.** *(Main, EqCo for negative pairs)* The mutual information lower bound of InfoNCE loss in Eq. 1 is irrelevant to the number of negative pairs $K$, if

$$
m = \tau\log\frac{\alpha}{K},
\tag{5}
$$

where $\alpha$ is a constant coefficient. And in the circumstances the bound is given by:

$$
f_{\text{bound}}\left(\tau\log\frac{\alpha}{K}, K\right) \approx \log(1 + \alpha) - \mathop{\mathbb{E}}_{\mathbf{q}\sim\mathcal{D},\mathbf{k}_0\sim\mathcal{D}'(\mathbf{q})} \log\left(1 + \alpha\frac{P(\mathbf{k}_0)}{P(\mathbf{k}_0|\mathbf{q})}\right) \approx f_{\text{bound}}(0, \alpha),
\tag{6}
$$

which can be immediately obtained by substituting Eq. 5 into Eq. 4. We name Eq. 5 as "*equivalent condition*".

Theorem 1 suggests a property of *equivalency*: under the condition of Eq. 5, no matter what the number of *physical* negative pairs $K$ is, the optimal solution of $\mathcal{L}_{NCE}$ (Eq. 1) is "equivalent" in the sense of the same mutual information lower bound. The bound is controlled by a hyper-parameter $\alpha$ rather than $K$. Eq. 6 further implies that the lower bound also correlates to the configuration of $K = \alpha$ without *margin*, which suggests we can "mimic" the InfoNCE loss's behavior of $K = K_0$ under a different physical negative sample size $K_1$, just by applying Eq. 5 with $\alpha = K_0$. It inspires us to simplify the existing state-of-the-art frameworks (e.g. MoCo (He et al. (2020))) with fewer negative samples but as accurate as the original configurations, which will be introduced next.

We empirically validate Theorem 1 as follows. Notice that $f_{\text{bound}}$ is difficult to calculate directly because $\mathcal{L}_{opt}$ is not known. Instead, we plot the *empirical mutual information lower bound*

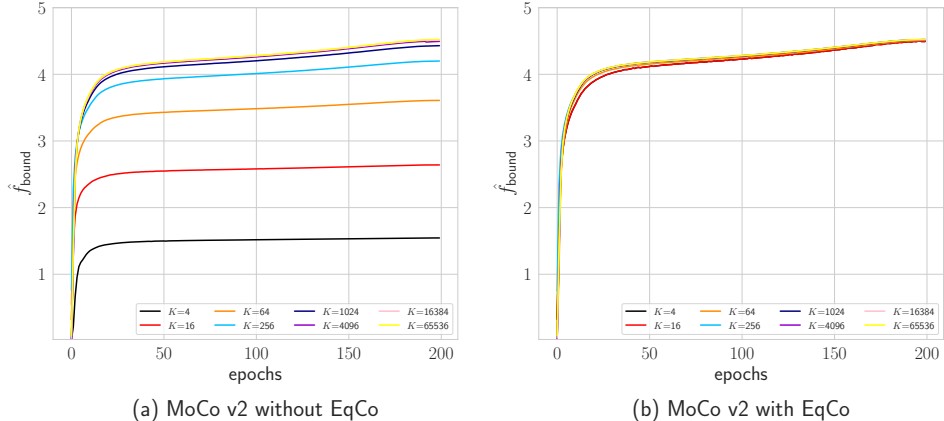

(a) MoCo v2 without EqCo  (b) MoCo v2 with EqCo

Figure 1: Evolution of the *empirical mutual information lower bound* $\hat{f}_{\text{bound}}$ during training. We use $\alpha = 65536$ for *EqCo*. Results are evaluated with *MoCo v2* on ImageNet. Refer to Theorem 1 for details. Best viewed in color.

$\hat{f}_{\text{bound}}(m, K) \triangleq \log(1 + Ke^{m/\tau}) - \mathcal{L}_{NCE}$. So, we have $\hat{f}_{\text{bound}} \leq f_{\text{bound}}$; when $\mathcal{L}_{NCE}$ converges to the optimum $\mathcal{L}_{opt}$, $\hat{f}_{\text{bound}}$ is an approximation of $f_{\text{bound}}$. In Fig. 1, we plot the evolution of $\hat{f}_{\text{bound}}$ during the training of *MoCo v2* under different configurations. Obviously, when it converges, without EqCo $\hat{f}_{\text{bound}}$ keeps increasing with the number of negative pairs $K$; in contrast, after applying the *equivalent condition* (Eq. 5) $\hat{f}_{\text{bound}}$ converges to almost the same value under different $K$s. The empirical results are thus consistent with Theorem 1.

**Remarks 1.** The equivalent condition in Eq. 5 suggests the margin $m$ is inversely correlated with $K$. It is intuitive, because the larger $K$ is, the more risks of *class collision* (Arora et al. (2019)) it suffers from, so we need to avoid over-penalty for negative samples near the query, thus smaller $m$ is used; in contrast, if $K$ is very small, we use larger $m$ to exploit more "hard" negative samples.

Besides, recall that the *margin* term $e^{m/\tau}$ is defined as the ratio of the prior probabilities $P^-/P^+$ in Eq. 2. If the equivalent condition Eq. 5 satisfies, i.e., $P^-/P^+ = \alpha/K$, we have $P^+ = 1/(1 + \alpha)$ (notice that $KP^- + P^+ \equiv 1$), suggesting that the prior probability of the ground truth distribution $H_0$ is supposed to be a constant ignoring the number of negative samples $K$. While in previous works (usually without the *margin* term, or $m = 0$) we have $P^+ = 1/(K + 1)$. It is hard to distinguish which prior is more reasonable. However at least, we intuitively suppose keeping a constant prior for the ground truth distribution may help to keep the optimal choices of hyper-parameters steady under different $K$s, which is also consistent with our empirical observations.

**Remarks 2.** In Theorem 1, it is worth noting that $K$ refers to the number of negative samples *per query*. In the conventional batched training scheme, negative samples for different queries could be either (fully or partially) shared or isolated, i.e., the total number of distinguishing negatives samples *per batch* could be different, which is not ruled by Theorem 1. However, we empirically find the differences in implementation do not result in much of the performance variation.

The following theorem further supports the equivalent rule (Theorem 1) from back-propagation view:

**Theorem 2.** Given the *equivalent condition* (Eq. 5) and a query embedding $\mathbf{q}$ as well as the corresponding positive sample $\mathbf{k}_0$, for $\mathcal{L}_{NCE}$ in Eq. 1 the expectation of the gradient norm w.r.t. $\mathbf{q}$ is bounded by [2]:

$$\mathbb{E}_{\mathbf{k}_i \sim \mathcal{D}'} \left\| \frac{d\mathcal{L}_{NCE}}{d\mathbf{q}} \right\| \leq \frac{2}{\tau} \left( 1 - \frac{\exp(\mathbf{q}^\top \mathbf{k}_0/\tau)}{\exp(\mathbf{q}^\top \mathbf{k}_0/\tau) + \alpha \mathbb{E}_{\mathbf{k}_i \sim \mathcal{D}'}[\exp(\mathbf{q}^\top \mathbf{k}_i/\tau)]} \right). \tag{7}$$

---

[2]Some works (e.g., He et al. (2020)) only use $d\mathcal{L}_{NCE}/d\mathbf{q}$ for optimization. In contrast, other works (Chen et al. (2020a)) also involve $d\mathcal{L}_{NCE}/d\mathbf{k}_i$, $(i = 0, \ldots, K)$, which we will investigate in the future.

Please refer to the Appendix A.1 for the detailed proof. Note that we assume the embedding vectors are *normalized*, i.e., $\|\mathbf{k}_i\| = 1(i = 0, \cdots, K)$, which is also a convention in recent contrastive learning works.

Theorem 2 indicates that, equipped with the equivalent rule (Eq. 5), the upper bound of the gradient norm is irrelevant to the number of negative samples $K$. Fig. 4 (see the Appendix A.2) further validates our theory: the gradient norm becomes much more steady after using *EqCo* under different $K$s. Since the size of $K$ affects little on the gradient magnitude, gradient scaling techniques, e.g. *linear scaling rule*, are not required specifically for different $K$s. Eq. 7 also implies that the *temperature* $\tau$ significantly affects the gradient norm even EqCo is applied – it is why we only recommend to modify $m$ for equivalence (Eq. 5), though the mutual information lower bound is determined by $e^{m/\tau}$ as a whole.

## 2.2 THE CASE OF POSITIVE PAIRS

In practice the InfoNCE loss (Eq. 1) is usually optimized with *batched SGD*, which can be represented as *empirical risk minimization*:

$$\mathcal{L}_{NCE}^{\text{batch}} = \frac{1}{N} \sum_{j=1}^{N} \mathcal{L}_{NCE}^{(j)}(\mathbf{q}_j, \mathbf{k}_{j,0}), \tag{8}$$

where $N$ is the number of *queries* (or *positive pairs*) *per batch*; $(\mathbf{q}_j, \mathbf{k}_{j,0}) \sim (\mathcal{D}, \mathcal{D}'(\mathbf{q}_j))$ is the $j$-th positive pair, and $\mathcal{L}_{NCE}^{(j)}(\mathbf{q}_j, \mathbf{k}_{j,0})$ is the corresponding loss. For different $j$, $\mathcal{L}_{NCE}^{(j)}$ is (almost) independent of each other, because $\mathbf{q}_j$ is sampled independently. Hence, Eq. 8 satisfies the form of *linear scaling rule* (Krizhevsky (2014); Goyal et al. (2017)), suggesting that **the learning rate should be adjusted proportional to the number of queries $N$ per batch**.

**Remarks 3.** Previous work like *SimCLR* (Chen et al. (2020a)) also proposes to apply linear scaling rule. [3] The difference is, in SimCLR it does not clarify the concept of "batch size" refers to the number of queries or the number of keys. However in our paper, we explicitly point that the linear scaling rule needs to be applied corresponding to the number of queries per batch ($N$) rather than $K$.

## 2.3 EMPIRICAL EVALUATION

In this subsection we conduct experiments on the three state-of-the-art self-supervised contrastive learning frameworks – *MoCo* (He et al. (2020)), *MoCo v2* (Chen et al. (2020c)) and *SimCLR* (Chen et al. (2020a)) to verify our theory in Sec. 2.1 and Sec. 2.2. We propose to alter $K$ and $N$ separately to examine the correctness of our *equivalent rules*.

**Implementation details.** We follow most of the training and evaluation settings recommended in the original papers respectively. The only difference is, for SimCLR, we adopt *SGD with momentum* rather than *LARS* (You et al. (2017)) as the optimizer. We use *ResNet-50* (He et al. (2016)) as the default network architecture. 128-d features are employed for *query* and *key* embeddings. Unless specially mentioned, all models are trained on *ImageNet* (Deng et al. (2009)) for 200 epochs without using the ground truth labels. We report the top-1 accuracy under the conventional *linear evaluation protocol* according to the original paper respectively. The number of queries per batch ($N$) is set to 256 by default. All models are trained with 8 GPUs.

It is worth noting the way we alter the number of negative samples $K$ independent of $N$ during training. For MoCo and MoCo v2, we simply need to set the size of the memory bank to $K$. Specially, if $K < N$, in the current batch the memory bank is actually composed of $K$ random *keys* sampled from the previous batch. While for SimCLR, if $K < N$ we random sample $K$ negative keys for

---

[3]In SimCLR, the authors find that *square-root* learning rate scaling is more desirable with *LARS* optimizer (You et al. (2017)), rather than *linear scaling rule*. Also, their experiments suggest that the performance gap between large and small batch sizes become smaller under that configuration. We point that the direction is orthogonal to our equivalent rule. Besides, SimCLR does not explore the case of very small $K$s (e.g. $K <= 128$).

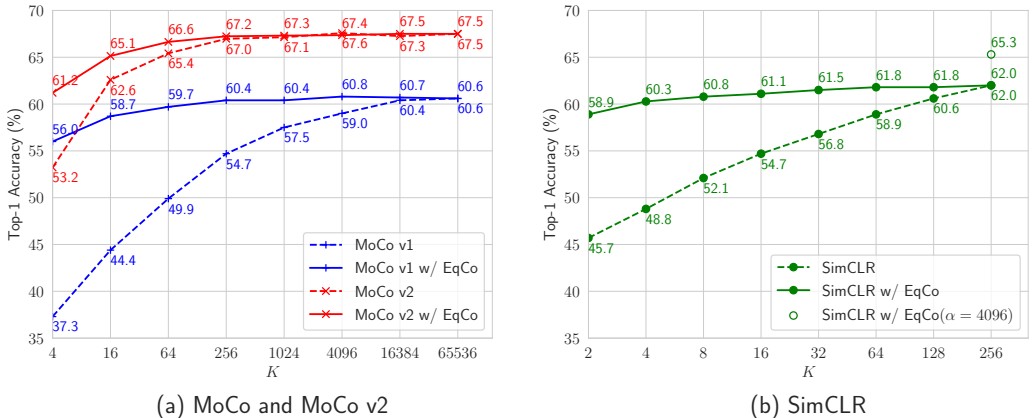

Figure 2: Comparisons with/without *EqCo* under different number of negative samples (noted by $K$). Results are evaluated with ImageNet top-1 accuracy using *linear evaluation protocol*. In EqCo, we set $\alpha = 65536$ for MoCo and MoCo v2, and $\alpha = 256$ for SimCLR (except for one data point with $\alpha = 4096$, as noted in the legend). Best viewed in color.

each query independently. We do not study the case that $K > N$ for SimCLR. We mainly consider the ease of implementation in designing the strategies; as mentioned in *Remarks 2* (Sec. 2.1), it does not affect the empirical conclusion.

**Quantitative results.** Fig. 2 illustrates the effect of our *equivalent rule* under different $K$s. Our experiments start with the best configurations (i.e. $K = 65536$ for MoCo and MoCo v2, and $K = 256$ for SimCLR[4]), then we gradually reduce $K$ and benchmark the performance. Results in Fig. 2 indicates that, without EqCo the accuracy significantly drops if $K$ becomes very small (e.g. $K < 64$). While with EqCo, by setting $\alpha$ to "mimic" the optimal $K$, the performance surprisingly keeps steady under a wide range of $K$s. Fig. 2(b) further shows that in SimCLR, by setting $\alpha$ to a number larger than the physical batch size (e.g. 4096 *vs.* 256), the accuracy significantly improves from 62.0% to 65.3%, [5] suggesting the benefit of EqCo especially when the memory is limited. The comparison fully demonstrates EqCo is essential especially when the number of negative pairs is small.

Besides, Table 1 compares the results of *MoCo v2* under different number of queries $N$, while $K = 65536$ is fixed. It is clear that, with *linear scaling rule* (Krizhevsky (2014); Goyal et al. (2017)), the final performance is almost unchanged under different $N$, suggesting the effectiveness of our equivalent rule for $N$.

| $N(K = 65536)$ | 256 | 512 | 1024 |
|---|---|---|---|
| Top-1 accuracy (%) | 67.5 | 67.5 | 67.4 |

Table 1: ImageNet accuracy (MoCo v2) vs. the number of queries per batch ($N$). The learning rates during training are adjusted with *linear scaling rule*.

## 3 SiMo: a Simpler but Stronger Baseline

*EqCo* inspires us to rethink the design of contrastive learning frameworks. The previous state-of-the-arts like *MoCo* and *SimCLR* heavily rely on large quantities of negative pairs to obtain high

---

[4]In the original paper of SimCLR (Chen et al. (2020a)), the best number of negative pairs is around 4096. However, the largest $K$ we can use in our experiment is 256 due to GPU memory limit.

[5]Our "mimicking" result (65.3%, $\alpha = 4096$, $K = 256$) is slightly lower than the counterpart score reported in the original SimCLR paper (66.6%, with a physical batch size of $K = 4096$), which we think may be resulted from the extra benefits of *SyncBN* along with *LARS* optimizer used in SimCLR, especially when the physical batch size is large.

performances, hence implementation tricks such as memory bank and large batch training are introduced, which makes the system complex and tends to be costly. Thanks to EqCo, we are able to design a simpler contrastive learning framework with fewer negative pairs.

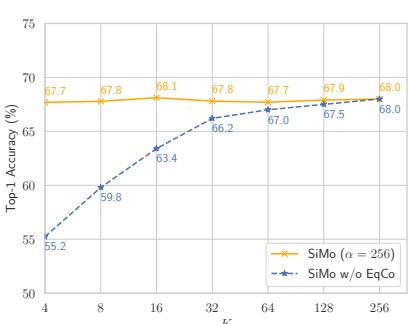

Figure 3: SiMo with/without EqCo

| Method | Epochs | Top-1 (%) |
|---|---|---|
| CPC v2 (Hénaff et al., 2019) | 200 | 63.8 |
| CMC (Tian et al., 2019) | 240 | 66.2 |
| SimCLR (Chen et al., 2020a) | 200 | 66.6 |
| MoCo v2 (Chen et al., 2020c) | 200 | 67.5 |
| InfoMin Aug. (Tian et al. (2020)) | 200 | **70.1** |
| SiMo ($K = 16, \alpha = 256$) | 200 | 68.1 |
| SiMo ($K = 256, \alpha = 256$) | 200 | 68.0 |
| SiMo ($K = 256, \alpha = 65536$) | 200 | 68.5 |
| PIRL (Misra & Maaten, 2020) | 800 | 63.6 |
| SimCLR (Chen et al., 2020a) | 1000 | 69.3 |
| MoCo v2 (Chen et al., 2020c) | 800 | 71.1 |
| InfoMin Aug. (Tian et al. (2020)) | 800 | **73.0** |
| SiMo ($K = 256, \alpha = 256$) | 800 | 71.8 |
| SiMo ($K = 256, \alpha = 65536$) | 800 | 72.1 |

Table 2: State-of-the-art InfoNCE-based frameworks

We propose *SiMo*, a simplified variant of *MoCo v2* (Chen et al. (2020c)) equipped with EqCo. We follow most of the design in Chen et al. (2020c), where the key differences are as follows:

**Memory bank.** MoCo, MoCo v2 and SimCLR v2 [6] (Chen et al. (2020b)) employ memory bank to maintain large number of negative embeddings $k_i$, in which there is a side effect: every positive embedding $k_0$ is always extracted from a "newer" network than the negatives' in the same batch, which could harm the performance. In SiMo, we thus cancel the memory bank as we only rely on a few negative samples per batch. Instead, we use the *momentum encoder* to extract both positive and negative *key* embeddings from the current batch.

**Shuffling BN vs. Sync BN.** In MoCo v1/v2, shuffling BN (He et al. (2020)) is proposed to remove the obvious dissimilarities of the BN (Ioffe & Szegedy (2015)) statistics between the positive (from current mini-batch) and the negatives (from memory bank), so that the model can make predictions based on the semantic information of images rather than the BN statistics. In contrast, since the positive and negatives are from the same batch in SiMo, therefore, we use sync BN (Peng et al. (2018)) for simplicity and more stable statistics. Sync BN is also used in SimCLR (Chen et al. (2020a)) and SimCLR v2 (Chen et al. (2020b)).

There are a few other differences, including 1) we use a BN attached to each of the fully-connected layers; 2) we introduce a warm-up stage at the beginning of the training, which follows the methodology in SimCLR (Chen et al. (2020a)). Apart from all the differences mentioned above, the architecture and the training (including data augmentations) details in SiMo are *exactly* the same as MoCo v2's. In the following text, the number of queries per batch (N) is set to 256, and the backbone network is *ResNet-50* by default.

**Quantitative results.** First, we empirically demonstrate the necessity of *EqCo* in SiMo framework. We choose the number of negative samples $K = 256$ as the baseline, then reduce $K$ to evaluate the performance. Fig. 3 shows the result on ImageNet using *linear evaluation protocol*. Without EqCo, the accuracy significantly drops when $K$ is very small. In contrast, using EqCo to "mimic" the case of large $K$ (by setting $\alpha$ to 256), the accuracy almost keeps steady even under very small $K$s.

Table 2 further compares our SiMo with state-of-the-art self-supervised contrastive learning methods on ImageNet. [7] Using only 16 negative samples per query, SiMo outperforms MoCo v2 (68.1% vs. 67.5%). If we increase $\alpha$ to 65536 to "simulate" the case under huge number of negative pairs, the accuracy further increases to 68.5%. Moreover, when we extend the training epochs to 800, we get the accuracy of 72.1%, surpassing the baseline MoCo v2 by 1.0%. The only entry that surpasses

---

[6] SimCLR v2 compares the settings with/without memory bank. However, they suggest employing memory bank as the best configuration.

[7] We mainly compare the methods with InfoNCE loss (Eq. 1) here, though recently *BYOL* (Grill et al. (2020)) and *SWAV* achieve better results using different loss functions.

our results is *InfoMin Aug.* (Tian et al. (2020)), which is mainly focuses on data generation and orthogonal to ours. The experiments indicate that SiMo is a simpler but more powerful baseline for self-supervised contrastive learning. Readers can refer to the Appendix B for more experimental results of SiMo.

## 4 LIMITATIONS AND FUTURE WORK

Theorem 1 suggests that given the *equivalent condition* (Eq. 5), InfoNCE losses under various $K$s are "equivalent" in the sense of the same mutual information lower bound, which is also backed up with the experiments in Fig. 1. However, Fig. 2 (a) shows that if $K$ is smaller than a certain value (e.g. $K \leq 16$), some frameworks like *MoCo v2* start to degrade significantly even with EqCo; while for other frameworks like *SiMo* (Fig. 3), the accuracy almost keeps steady for very small $K$s. Tschannen et al. (2019) also point that the principle of *InfoMax* cannot explain all the phenomena in contrastive learning. We will investigate the problem in the future, e.g. from other viewpoints such as gradient noise brought by small $K$s (Fig. 4 in Appendix A.2 gives some insights).

Though the formulation of Eq. 1 is very common in the field of supervised *metric learning*, which is usually named *margin softmax cross-entropy loss* (Deng et al., 2019; Wang et al., 2018; Sun et al., 2020). Nevertheless, unfortunately, our equivalent rule seems invalid to be generalized to those problems (e.g. face recognition). The major issue lies in the approximation in Eq. 3, we need the negative samples $\mathbf{k}_i$ to be independent of the query $\mathbf{q}$, which is not satisfied in supervised tasks.

According to Fig. 2 and Fig. 3, the benefits of EqCo become significant if $K$ is sufficiently small (e.g. $K < 64$). But in practice, for modern computing devices (e.g. GPUs) it is not that difficult to use $\sim 256$ negative pairs per query. Applying EqCo to "simulate" more negative pairs via adjusting $\alpha$ can further boost the performance, however, whose accuracy gains become relatively marginal. For example, in Table 2 under 200 epochs training, SiMo with $\alpha = 65536$ outperforms that of $\alpha = 256$ by only 0.5%. It could be a fundamental limitation of InfoNCE loss. We will investigate the problem in the future.

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

## A    DETAILS ABOUT **THEOREM 2**

### A.1    PROOF OF EQ. 7

Given the *equivalent condition* (Eq. 5) and a query embedding $\mathbf{q}$ as well as the corresponding positive sample $\mathbf{k}_0$, for $\mathcal{L}_{NCE}$ in Eq. 1 the expectation of the gradient norm w.r.t. $\mathbf{q}$ is bounded by:

$$\mathop{\mathbb{E}}_{\mathbf{k}_i \sim \mathcal{D}'} \left\| \frac{d\mathcal{L}_{NCE}}{d\mathbf{q}} \right\| \leq \frac{2}{\tau} \left( 1 - \frac{\exp(\mathbf{q}^\top \mathbf{k}_0/\tau)}{\exp(\mathbf{q}^\top \mathbf{k}_0/\tau) + \alpha \mathbb{E}_{\mathbf{k}_i \sim \mathcal{D}'}[\exp(\mathbf{q}^\top \mathbf{k}_i/\tau)]} \right). \tag{9}$$

*Proof.* For simplicity, we denote the term $\exp(\mathbf{q}^\top \mathbf{k}_i/\tau)$ as $s_i (i = 0, \dots, K)$. Then $\mathcal{L}_{NCE}$ can be rewritten as:

$$\mathcal{L}_{NCE} = -\log \frac{s_0}{s_0 + \frac{\alpha}{K} \sum_{i=1}^{K} s_i} \tag{10}$$

The gradient of $\mathcal{L}_{NCE}$ with respect to $\mathbf{q}$ is easily to derived:

$$\frac{d\mathcal{L}_{NCE}}{d\mathbf{q}} = -\frac{1}{\tau} \left( 1 - \frac{s_0}{s_0 + \frac{\alpha}{K} \sum_{i=1}^{K} s_i} \right) \mathbf{k}_0 + \frac{\alpha}{\tau K} \sum_{i=1}^{K} \frac{s_0}{s_0 + \frac{\alpha}{K} \sum_{i=1}^{K} s_i} \mathbf{k}_i, \tag{11}$$

Owing to the *Triangle Inequality* and the fact that $\mathbf{k}_i (i = 0, \dots, K)$ is normalized, the norm of gradient is bounded by:

$$\begin{aligned}
\left\| \frac{d\mathcal{L}_{NCE}}{d\mathbf{q}} \right\| &\leq \left| \frac{1}{\tau} \left( 1 - \frac{s_0}{s_0 + \frac{\alpha}{K} \sum_{i=1}^{K} s_i} \right) \right| \cdot \|\mathbf{k}_0\| + \sum_{i=1}^{K} \left| \frac{\alpha}{\tau K} \frac{s_i}{s_0 + \frac{\alpha}{K} \sum_{i=1}^{K} s_i} \right| \cdot \|\mathbf{k}_i\| \\
&= \frac{1}{\tau} \left( 1 - \frac{s_0}{s_0 + \frac{\alpha}{K} \sum_{i=1}^{K} s_i} \right) + \frac{1}{\tau} \sum_{i=1}^{K} \frac{\frac{\alpha}{K} s_i}{s_0 + \frac{\alpha}{K} \sum_{i=1}^{K} s_i} \\
&= \frac{2}{\tau} \left( 1 - \frac{s_0}{s_0 + \frac{\alpha}{K} \sum_{i=1}^{K} s_i} \right)
\end{aligned} \tag{12}$$

Since the cosine similarity between $\mathbf{q}$ and $\mathbf{k}_i$ $(i = 1, \dots, K)$ is bounded in $[-1, 1]$, we know the expectation of $\mathbb{E}_{\mathbf{k}_i \sim \mathcal{D}'}[s_i]$ exists. According to Inequality (12) and *Jensen's Inequality*, we have:

$$\begin{aligned}
\mathop{\mathbb{E}}_{\mathbf{k}_i \sim \mathcal{D}'} \left[ \frac{2}{\tau} \left( 1 - \frac{s_0}{s_0 + \frac{\alpha}{K} \sum_{i=1}^{K} s_i} \right) \right] &= \frac{2}{\tau} \left( 1 - \mathop{\mathbb{E}}_{\mathbf{k}_i \sim \mathcal{D}'} \left[ \frac{s_0}{s_0 + \frac{\alpha}{K} \sum_{i=1}^{K} s_i} \right] \right) \\
&\leq \frac{2}{\tau} \left( 1 - \frac{s_0}{s_0 + \alpha \mathbb{E}_{\mathbf{k}_i \sim \mathcal{D}'}[s_i]} \right)
\end{aligned} \tag{13}$$

Replacing $s_i$ by $\exp(\mathbf{q}^\top \mathbf{k}_i/\tau)$, the proof of Theorem 2 is completed.

$\square$

## A.2 EMPIRICAL EVALUATION ON THE MAGNITUDE OF GRADIENTS

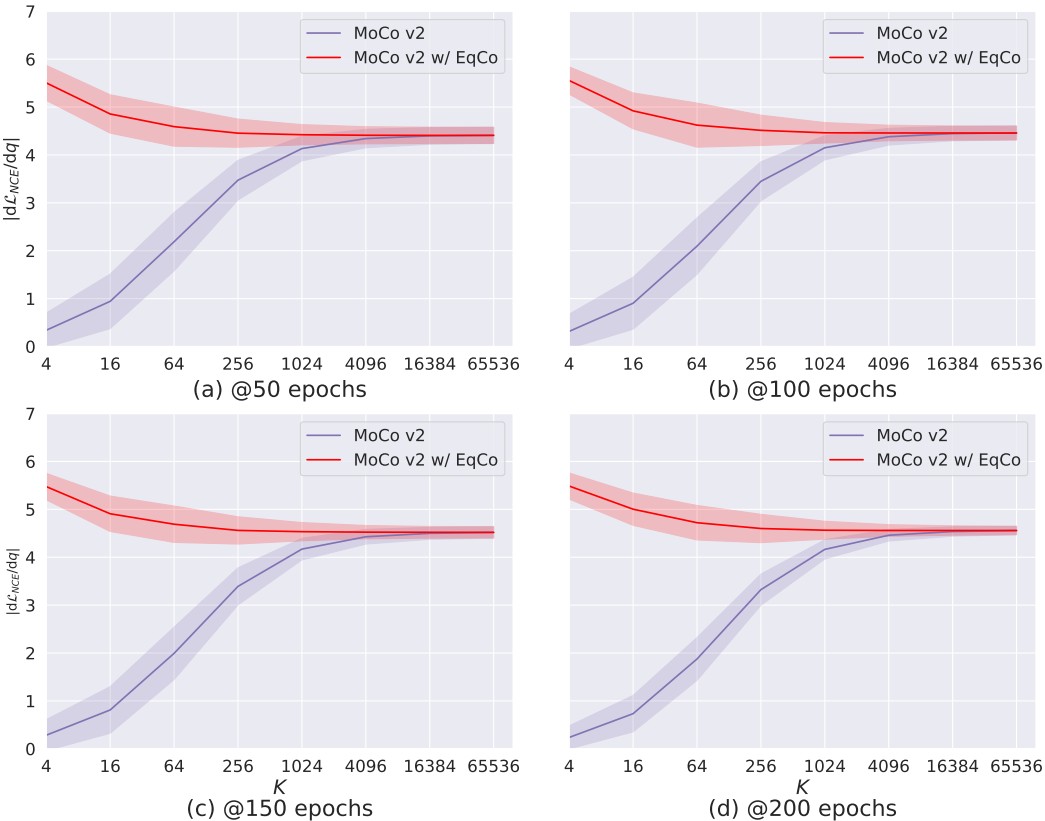

Figure 4: The means (solid line) and variances (ribbon, $\pm\sigma$) of $\|\mathrm{d}\mathcal{L}_{NCE}/\mathrm{d}\boldsymbol{q}\|$ under different $K$s. We train a normal MoCo v2 for 200 epochs and show the statistics at different epochs.

## B    MORE EXPERIMENTS ON SIMO

For the following experiments of this section, we report the top-1 accuracy of SiMo on ImageNet (Deng et al., 2009) under the linear evaluation protocol. The backbone of SiMo is ResNet-50 (He et al., 2016) and we train SiMo for 200 epochs unless noted otherwise.

### B.1    ABLATION ON MOMENTUM UPDATE

In MoCo (He et al., 2020) and MoCo v2 (Chen et al., 2020c), the key encoder is updated by the following rule:

$$\theta_k = \beta\theta_k + (1 - \beta)\,\theta_q$$

where $\theta_q$ and $\theta_k$ stand for the weights of query encoder and key encoder respectively, and $\beta$ is the momentum coefficient. For SiMo, we also adopt the momentum update and use the key encoder to compute the features of positive sample and negative samples.

In Table 3, we report the results of SiMo with different momentum coefficients. The number of training epochs is set to be 100, so the top-1 accuracy of baseline ($\beta = 0.999$) drops to 64.4%. Compared to the baseline, SiMo without momentum update ($\beta = 0$) is inferior, showing the advantage of momentum update.

| $\beta$ | 0 | 0.999 |
|---|---|---|
| Accuracy (%) | 62.1 | 64.4 |

Table 3: Ablation on momentum update.

### B.2    ABLATION ON BN

Table 4 shows the performance of SiMo equipped with shuffling BN or Sync BN. Likewise, we train SiMo for 100 epochs. It is easy to check out that SiMo with shuffling BN struggles to perform well. Besides, compared to MoCo v2, SiMo with shuffling BN degrades significantly, and we conjecture that it is because the MLP structure of SiMo is more suitable for Sync BN, rather than shuffling BN.

|  | Shuffling BN | Sync BN |
|---|---|---|
| Accuracy (%) | 58.8 | 64.4 |

Table 4: Sync BN vs. shuffling BN.

### B.3    SIMO WITH DIFFERENT $\alpha$

As shown in Sec.2.1, $\alpha$ is related to the lower bound of mutual information. Table 5 reveals how accuracy of SiMo varies with the choice of $\alpha$. As we increase $\alpha$ to 65536, the accuracy tends to improve, in accordance with the Eq.6. However, when $\alpha$ is too large (e.g., 262144), the performance slightly drops by 0.2%.

| $\alpha$ | 256 | 1024 | 4096 | 16384 | 65536 | 262144 |
|---|---|---|---|---|---|---|
| Accuracy (%) | 68.0 | 68.1 | 68.1 | 68.4 | 68.5 | 68.3 |

Table 5: SiMo with different $\alpha$.

Similar results can be found in MoCo v2. We increase $K$ to 262144 in MoCo v2, the accuracy also descends (in Table 6).

| $K$ | 256 | 1024 | 4096 | 16384 | 65536 | 262144 |
|---|---|---|---|---|---|---|
| Accuracy (%) | 67.0 | 67.1 | 67.6 | 67.3 | 67.5 | 67.4 |

Table 6: MoCo v2 with different $K$.

### B.4 SiMo with Wider Models

Results using wider models are presented in Table 7. For SiMo, the performance is further boosted with wider models (more channels). For instance, SiMo with ResNet-50 (2x) and ResNet-50 (4x) outperforms the baseline (68.5%) by 2% and 3.8% respectively.

| Architecture | Param. (M) | $\alpha$ | Top-1 (%) |
|---|---|---|---|
| ResNet-50 (2x) | 94 | 256 | 70.2 |
| ResNet-50 (2x) | 94 | 65536 | 70.5 |
| ResNet-50 (4x) | 375 | 256 | 71.9 |
| ResNet-50 (4x) | 375 | 65536 | 72.3 |

Table 7: SiMo with wider models. All models are trained with 200 epochs.

### B.5 Transfer to object detection

**Setup** We utilize FPN (Lin et al., 2017) with a stack of 4 $3 \times 3$ convolution layers in R-CNN head to validate the effectiveness of SiMo. Following the MoCo training protocol, we fine-tune with synchronized batch-normalization (Peng et al., 2018) across GPUs. The additional initialized layers are also equipped with BN for stable training. To effectively validate the transferability of the features, the training schedule is set to be 12 epochs (known as $1\times$), in which learning rate is initialized as 0.2 and decreased at 7 and 11 epochs with a factor of 0.1. The image scales are random sampled of $[640, 800]$ pixels during training and fixed with 800 at inference.

**Results** Table 8 summarizes the fine-tuning results on COCO val2017 of different pre-training methods. Random initialization indicates training COCO from scratch, and supervised represents conventional pre-training with ImageNet labels. Compared with MoCo, SiMo achieves competitive performance without large quantities of negative pairs. It is also on a par with the supervised counterpart and significantly outperforms random initialized one.

| pre-train | AP | $AP_{50}$ | $AP_{75}$ | $AP_s$ | $AP_m$ | $AP_l$ |
|---|---|---|---|---|---|---|
| random init | 31.4 | 49.4 | 34.0 | 17.9 | 32.3 | 41.6 |
| supervised | 39.0 | 59.1 | 42.6 | 22.4 | 42.2 | 50.6 |
| MoCo v2 | 39.1 | 59.2 | 42.5 | 23.3 | 42.1 | 50.8 |
| SiMo | 39.0 | 59.2 | 42.3 | 22.9 | 41.8 | 50.5 |

Table 8: Object detection fine-tuned on COCO.

## C A Toy Evaluation of EqCo

To evaluate the effectiveness of EqCo as mutual information (MI) estimator, following the configuration of Poole et al. (2019), we estimate the MI lower bound of between two simple random vectors.

Specifically, given that $(X, Y)$ are drawn from the known correlated Gaussian distribution, we calculate the lower bound of MI between $X$ and $Y$ based on their embedding. $X$ is a 20-dimensional random variables drawn from a standard Gaussian distribution. And we sampled $Y$ with the following rule:

$$Y = \rho X + \sqrt{1 - \rho^2}\,\epsilon \tag{14}$$

where $\rho$ is a the given correlation coefficient and $\epsilon$ is a random variable sampled from a standard Gaussian distribution and independent from $X$. With a known $\rho$, the ground truth MI between $X$ and $Y$ is easy to compute:

$$\mathcal{I}(X, Y) = -\frac{d}{2} \log \left(1 - \rho^2\right) \tag{15}$$

Here, $d$ is the dimension of $X$ and $Y$, and as mentioned above we set $d = 20$.

To embed $X$ and $Y$, we adopt two MLPs respectively, and each MLP has 1 hidden layer of 256 units, followed by ReLU activation function. We use Adam optimizer with learning rate of 0.0005 to optimize InfoNCE or EqCo for 5000 steps. For each training iteration, $K$ pairs of $(X, Y)$ are independently sampled, which means there are $K-1$ negative samples for each query. After training, the weights of MLPs are frozen and we repeat estimating the lower bound of MI for 1000 times to reduce the estimating variance. For experiments with EqCo, we set the $\alpha = 512$.

As shown in Table 9, $\mathcal{I}_{NCE}$ varies with $K$, while $\mathcal{I}_{EqCo}$ remains steady. Especially, when the ground truth MI is relatively large (e.g., 8, 10), significant differences between EqCo and InfoNCE can be observed. The experiment further validates the effectiveness of EqCo.

| | $K$ | | | |
| | 64 | 128 | 256 | 512 |
|---|---|---|---|---|
| *Mutual Information* $= 2.0$ | | | | |
| $\mathcal{I}_{NCE}$ | 1.7 | 1.8 | 1.9 | 1.9 |
| $\mathcal{I}_{EqCo}$ | 1.9 | 1.9 | 1.9 | 1.9 |
| *Mutual Information* $= 4.0$ | | | | |
| $\mathcal{I}_{NCE}$ | 2.9 | 3.2 | 3.4 | 3.6 |
| $\mathcal{I}_{EqCo}$ | 3.8 | 3.7 | 3.6 | 3.6 |
| *Mutual Information* $= 6.0$ | | | | |
| $\mathcal{I}_{NCE}$ | 3.6 | 4.1 | 4.5 | 4.9 |
| $\mathcal{I}_{EqCo}$ | 5.1 | 5.0 | 4.9 | 4.9 |
| *Mutual Information* $= 8.0$ | | | | |
| $\mathcal{I}_{NCE}$ | 3.9 | 4.6 | 5.1 | 5.6 |
| $\mathcal{I}_{EqCo}$ | 5.8 | 5.7 | 5.7 | 5.6 |
| *Mutual Information* $= 10.0$ | | | | |
| $\mathcal{I}_{NCE}$ | 4.1 | 4.7 | 5.4 | 6.0 |
| $\mathcal{I}_{EqCo}$ | 6.1 | 6.0 | 6.0 | 6.0 |

Table 9: Estimating mutual information by InfoNCE and EqCo with different batch size and various ground truth mutual information.

