# OpenReview forum: "EqCo:   Equivalent Rules for Self-supervised Contrastive Learning"
_ICLR.cc/2021/Conference — Reject_

### Official Review · AnonReviewer2 · 2020-10-23
**Excellent and Insightful Discussion on Negative Samples in Self-supervised Contrastive Learning**

**Rating:** 8
**Confidence:** 5

**Review:**

*****************************
Summary:

This paper focuses on self-supervised contrastive learning. Previous contrastive learning methods heavily rely on a large number of negative samples. This paper proposed a novel method with an additional margin term, and mathematically investigate the relationship among the margin term, the temperature, and the number of negative samples. The number of negative samples can be significantly reduced by tuning the margin term, while the performance remains more stable compared to previous contrastive learning methods. Furthermore, this paper proposed a MoCo-based strong baseline that can achieve comparable results with an extremely small number of negative samples.

*****************************
Reasons for Score:

Overall, I vote for acceptance. This paper proposed equivalent rules which successfully reduce the number of negative samples for contrastive representation learning while maintaining the performance. This motivation is very novel and interesting, and I believe the findings in this paper significantly contributes to the community of contrastive representation learning. My main concern is the assumption that the prior probabilities $P^+$ and $P^-$ satisfy $P^+/P^-=e^{-m/\tau}$ (see Sec. 2.1).

*****************************

Strengths:

\+ This paper focused on the problem of using a large number of negative samples in self-supervised contrastive learning, which is very important but not fully investigated before.

\+ This paper proposed a simple but strong baseline that can achieve comparable results with an extremely small number of negative samples. To the best of my knowledge, this is the first time to succeed with a small number of negative samples in contrastive learning. It may also inspire the applications in other self-supervised learning problems where a large number of negative samples are hard to be maintained.

\+ This paper provides detailed mathematical analysis and theorems, which makes the paper theoretically strong.

\+ The proposed strategy achieves significant improvement compared to the most popular self-supervised contrastive learning methods, MoCo and SimCLR, with a small number of negative samples.

*****************************

Concerns & Questions:

\- The Eq. (2) includes $P^+$ and $P^-$ as prior probabilities. It is not clear to me how the priors are taken into account and whether it is theoretically correct. In addition, it is not clear the assumption $P^+/P^−=e^{−m/\tau}$ is correct. Could more mathematical details be introduced? Why is including $P^+$ and $P^-$ called a generalized form?

\* In this paper, the temperature $\tau$ is fixed when tuning $K$ and selecting $m$. It would be an interesting issue if Eq. (5) is rewritten as $\tau=\frac{m}{\log\alpha-\log K}$. I wonder whether it is possible to select $\tau$ by fixing $m$ and tuning $K$. If not, is there any insight behind this?

\* In the proposed SiMo, all the negative examples are obtained from the current batch via the momentum encoder rather than the dictionary. Table 3 in the appendix shows that momentum update is important. Note that the momentum encoder in MoCo is used for the consistent dictionary of negative samples. Since the dictionary is not used in SiMo, the momentum update should have a different role in contrastive learning, perhaps like Mean Teacher. It would be great if more analysis or explanation on this can be provided.

\* SiMo uses Sync BN rather than Shuffling BN. I wonder whether Sync BN causes extra computational cost or slowers the training speed.





============================================


After rebuttal:

According to the reviewers' feedback, I would keep my score to 8 and still vote for acceptance. However, there are still two details that are expected to be fixed in the future version. First, the reason why momentum update in SiMo is important is not convincing to me. It is not clear why letting $\theta_k=\theta_q$ cannot ensure the loss becoming smaller. More theoretical and experimental analyses are expected to address this issue. I still encourage the authors to rethink this detail. Second, the computational cost is provided in the feedback. I encourage the authors to include the numbers in the paper. 35\% additional cost cannot be ignored.

---

> ### Author Response · Authors · 2020-11-24
> **Response to AnonReviewer2**
>
> Thanks for the review comments.
>
> Q1: It is not clear to me how the priors $P^+$ and $P^-$ are taken into account and whether it is theoretically correct. In addition, it is not clear the assumption $P^+/P^-=e^{-m/\tau}$ is correct. Could more mathematical details be introduced? Why is including $P^+$ and $P^-$ called a generalized form?
>
> A1: We try to explain the priors as the following viewpoints:
>
> First, our derivation is mainly inspired by CPC (Oord et al. (2018)). CPC does not introduce the priors explicitly, but from the Bayesian view, there must exists a prior probability $\text{P}(H_i)$ for each candidate distribution $H_i$. CPC implicitly set those priors to be equal, however, which is not demonstrated to be correct. So, in the paper, we explicitly formulate $P^+$ and $P^-$ to take them into account, which is called the generalized form of CPC.
>
> Second, we point that $P^+/P^-=e^{-m/\tau}$ is not an assumption but a definition, which is parameterized by a variable $m$; therefore, the formulation is always "correct". However, the value of $m$ is still difficult to be calculated. We further assume that $m$ is a function of $K$, and the "correct value" of the $(m, K)$ pair corresponds to the best performance. So, our equivalent rule in Eq.(6) can be interpreted as follows: first, we fix the value of $m$ (e.g. $m=0$) and explore the $K$ with best performance (noted as $K=\alpha$), in which we assume the pair $(m=0, K=\alpha)$ is "correct"; then, based on the equivalent condition  that MI lower bound should be unchanged, we can predict the "correct" value of $m$ under other number of negative samples $K'$.
>
>
> Q2: In this paper, the temperature $\tau$ is fixed when tuning $K$ and selecting $m$. It would be an interesting issue if Eq. (5) is rewritten as $\tau=\frac{m}{\text{log}\ \alpha - \text{log}\ K}$. I wonder whether it is possible to select $\tau$  by fixing $m$ and tuning $K$. If not, is there any insight behind this?
>
> A2: As we explained in the paragraph above Sec 2.2, $\tau$ affects not only the MI lower bound but also the gradient (Eq. 7). So, to get equivalent at least learning rate also needs to be tuned. Our preliminary study shows that it seems to work under some configurations of $(m, K)$, but not in general.
>
>
> Q3: In the proposed SiMo, all the negative examples are obtained from the current batch via the momentum encoder rather than the dictionary. Table 3 in the appendix shows that momentum update is important. Note that the momentum encoder in MoCo is used for the consistent dictionary of negative samples. Since the dictionary is not used in SiMo, the momentum update should have a different role in contrastive learning, perhaps like Mean Teacher. It would be great if more analysis or explanation on this can be provided.
>
> A3: The reason why momentum update in SiMo still matters is that we only update the network with the partial gradient subject to the parameters in the query encoder, which follows the convention in MoCo. Specially, the loss in SiMo can be formulated as follows:
>
> $$
> L=L(q(\theta_q), k(\theta_k))
> $$
>
> And we update the parameters using the partial gradient:
>
> $$
> \nabla_{\theta_q} L = \frac{\partial L}{\partial q} \frac{\partial q}{\partial \theta_q}
> $$
>
> If we do not use momentum update, i.e. simply letting $\theta_k = \theta_q$, we cannot ensure that for each iteration the loss becomes smaller only using $\nabla_{\theta_q} L$, because another term $\nabla_{\theta_k} L$ is not taken into account. However, if we introduce momentum update, the update of $\theta_k$ thus becomes very slow, so for each iteration we can roughly regard $\theta_k$ as a constant. So, the full gradient is only determined by $\nabla_{\theta_q} L$, thus our optimization becomes correct. Furthermore, if $\theta_k$ is set to be learnable, it is equivalent to SimCLR, which already shows competitive performance.
>
> Q4: Whether Sync BN causes extra computational cost or slower the training speed.
>
> A4: It just depends on the implementation and the computing devices. By using the Sync BN implemented in Detectron, the additional cost is around 45\%, while built-in SyncBatchNorm of PyTorch only requires about 35\% additional cost. It can be further accelerated with GPUs that support P2P access, e.g. V100.

---

### Official Review · AnonReviewer3 · 2020-10-26
**Interesting topic and observation, but there are some issues with the theory and result**

**Rating:** 5
**Confidence:** 5

**Review:**

This paper studies the performance of contrastive learning under a varying number of negative examples. The authors propose a small modification to the InfoNCE loss and show tuning some hyper-parameter could significantly improve the results with a small number of negatives. I found this topic very interesting, and the proposed solution to be quite simple. It challenges the conventional belief that a large number of negatives are required for contrastive learning (though this has been challenged before, recently by BYOL). The results show that when the number of negatives (K) is small, by using the proposed method, it could greatly improve the performance.

However, there are some questions/issues that I came across:

The theoretical analysis shows by setting m using eq 5, the mutual information lower bound could be made completely irrelevant to K; instead it’s controlled by a constant alpha. This is clearly problematic: 1) it suggests one could simply adjust alpha to improve the lower bound to an arbitrarily large number, which is impossible given that mutual information is a fixed number given a dataset. 2) experimental results show there is still a significant gap when using small K (Figure 2). This suggests alpha is not some number one could arbitrary set to improve the bound, and it may actually be related to K, which makes the current theoretical analysis potentially flawed or incomplete.

The experimental results are pretty nice when K is small (compared to baselines), however, these are somewhat inconsequent since a small K is obtained by dropping out some negative examples in the current or past mini-batch, not actual small batch size (one still cannot use a small batch size due to batch norm and actual training time). The improvement on the best of MoCov2 result looks somewhat marginal and may not be related to the number of negatives (which contradicts the claim on why SiMo helps), as the performance of MoCov2 has saturated with increased K (Table 6). If the author’s claim were true, it should be possible to improve SimCLR's performance on small batch size (to reach the same performance of large batch size training), but this was not shown in the paper (Figure 2b). What's more, Figure 2b looks worse than expected. According to SimCLR paper (Table B.1), the gap between small and big batch sizes can be significantly reduced by using a proper learning rate, so it would be more convincing to use a better learning rate (or compare performances with different learning rates).

Other minor issues: 1) for Table 7, the number of training epochs is not specified. 3) the methods in Table 2 looks a bit arbitrary, several existing similarly/better performing methods (e.g. BYOL, SWAV) are not included.

Given these issues, I will give an initial rating of 5, which is contingent on the responses from the authors.

--after rebuttal--

I appreciate the response, and it addressed my initial concern on the mutual information bound. However, the major concern remains which is the toy-ish setup and its practical value. The main result authors showed is that dropping some negatives doesn't hurt the performance if one adjusts m using EqCo rule. While this is interesting, the current setting of few negatives (by throwing out available negatives) is toy-ish, and may not reflect the real situation when a mini-batch is small thus only few negatives are available. I'd really like to see how this could be used to deal with the situation where you indeed only has few negatives. In MoCo, one could always easily buffer negatives with EMA the network, so there's no need to use a smaller set of negatives. In SimCLR, there seems to be a bigger potential of improvement from EqCo with actual smaller batch size, but the authors should also tune the learning rate (for smaller batch sizes) to make the baseline convincing, and compare to the results where large batch size is used. While the authors added a new point of alpha=4096, the current results of the paper are still incomplete, so I would keep my score. I'd encourage the authors to update/complete these results regardless the paper is immediately accepted by ICLR.

---

> ### Author Response · Authors · 2020-11-24
> **Response to AnonReviewer3**
>
> Thanks for the review comments.
>
> Q1. The theoretical analysis is clearly problematic: it suggests one could simply adjust alpha to improve the lower bound to an arbitrarily large number, which is impossible given that mutual information (MI) is a fixed number given a dataset.
>
> A1. Not really. In Eq.(6), for simplicity, we assume
> $\frac{\text{P}(\mathbf{k}_0)}{\text{P}(\mathbf{k}_0|\mathbf{q})} = c \quad (0<c<1)$ is a constant. Thus the bound becomes:
>
> $$
> f_{\text{bound}}(0, \alpha) \approx \text{log} (1 + \alpha) - \text{log} (1 + \alpha c)
> = \text{log} \left(\frac{1/\alpha + 1}{1/\alpha + c}\right),
> $$
>
> which monotonously increases with $\alpha$; when $\alpha \to +\infty$, the limit is $-\text{log} (c)$, which cannot be "arbitrarily large number". Experiments in Appendix C further validates that our modified loss can estimate MI as conventional InfoNCE loss.
>
> Q2. Experimental results show there is still a significant gap when using small K (Figure 2). This suggests alpha is not some number one could arbitrary set to improve the bound. The improvement on the best of MoCo v2 result looks somewhat marginal and may not be related to the number of negatives (which contradicts the claim on why SiMo helps). The improvement on the best of MoCov2 result looks somewhat marginal.
>
> A2. We acknowledge it correlates to a limitation of our method, as discussed in Sec.4 (first paragraph). The problem is, both Theorem 1 and Theorem 2 indicate the equivalence property in the sense of expectation, however, the variance indeed becomes large if $K$ is small while $\alpha$ is large. For example, in Eq.(13) the variance of the approximation clearly increases with large $\alpha$ and small $K$. So if $K$ is too small, the gradient noise could be large, which may lead to inferior results. However, it dose not mean our theoretical analysis is "clearly problematic". Our experiments in Fig. 2, Fig. 3 also show that the equivalent rule can still work within a wide range of $\alpha$ and $K$s. Furthermore, in SiMo and SimCLR (using SyncBN), equipped with EqCo we find the degradation of small $K$ is much smaller than that in MoCo series (using Shuffling BN), which may be because SyncBN helps to reduce the gradient noise.
>
> Q3. It should be possible to improve SimCLR's performance on small batch size (to reach the same performance of large batch size training), but this was not shown in the paper (Figure 2b). What's more, Figure 2b looks worse than expected.
>
> A3. In the revised version we introduce a new experiment using $\alpha = 4096$ and $K=256$ in SimCLR (see Fig.2(b)), which brings significant improvements (from 62.0 to 65.3). Though the result is slightly worse than that reported in SimCLR paper (66.6, with a physical batch size of 4096), we argue that the implementation may be not fully aligned (see our comments in Footnote 4 and 5). We are pleased to provide the fair comparison after our distributed SimCLR implementation is ready. In addition, we point that our reproduction of SimCLR under small batch size ($K=256$) is consistent with the counterpart in SimCLR paper (62.0 vs 61.9), not "worse than expected".
>
> Q4. According to SimCLR paper (Table B.1), the gap between small and big batch sizes can be significantly reduced by using a proper learning rate, so it would be more convincing to use a better learning rate.
>
> A4. We argue that EqCo is applicable but not the only way to bridge the gap between small and large batches. We agree that carefully tuning the learning rate may also help; but from the experiments in SimCLR paper, the gap seems still significant, and results under smaller batch sizes ($K<256$) is are not provided. We comment it in Footnote 3. We are pleased to add the related experiments afterwards.
>
> Q5. Other minor issues.
>
> A5. We have fixed them in the revised version. See Table 2, Footnote 7 and Table 7.

---

### Official Review · AnonReviewer1 · 2020-10-28
**Good paper with some concerns**

**Rating:** 6
**Confidence:** 4

**Review:**

The authors propose Eqco which can get rid of the effect of the large negative sample size in contrastive learning.

Pros:
1. The authors tackle an important question whether the large negative sample size is important in contrastive learning
2. The authors propose Eqco which introduces an margin term to the InfoNCE loss.
3. The authors empirically shows that the result algorithm Simo is less sensitive to negative sample size.
4. The authors prove that large negative sample size does not affect the lower bound and the bound of the gradient.
5. The authors improve Moco-v2's result using Eqco.

Cons:
1. In figure 2, even though the accuracy is less sensitive to K. But increasing K still helps the performance. This result is against the theorem 1 and 2. Is there any other explanation?
2. Gradient and lower bound may not be the only thing that affect the performance. What about generalization? Will large negative sample size help generalization? This needs a discussion
3. Does representation learning through minimizing the mutual information help improve the performance of the downstream classifier (linear or MLP)? Will large negative sample size help because of the downstream classifier is linear/MLP? This needs a discussion.

Significance:
The problem is significant and the authors made some progress on it.

Clarity:
The paper is clearly written in general.

Originality:
The paper is novel.

---

> ### Author Response · Authors · 2020-11-24
> **Response to AnonReviewer1**
>
> Thanks for the review comments.
>
> Q1. In figure 2, even though the accuracy is less sensitive to K, but increasing K still helps the performance. This result is against the theorem 1 and 2. Is there any other explanation?
>
> A1. Please refer to our response for AnonReviewer3 (Q2) and Sec.4 (first paragraph) for details. In short, it is because both Theorem 1 and Theorem 2 indicate the equivalence property in the sense of expectation, however, the variance indeed becomes large if $K$ is small while $\alpha$ is large.
>
>
> Q2. Gradient and lower bound may not be the only thing that affect the performance. What about generalization? Will large negative sample size help generalization? This needs a discussion.
>
> A2. We agree it is an interesting topic to theoretically analyze the generation bound under EqCo, which we will investigate in the future. But empirically, Table 8 shows that SiMo ($K=256, \alpha=65536$) achieves similar performance as MoCo v2 ($K=65536$) on COCO detection task, which may indicate that generation may not be a severe problem even though the physical negative batch size is small under EqCo.
>
> Q3. Does representation learning through minimizing the mutual information help improve the performance of the downstream classifier (linear or MLP)? Will large negative sample size help because of the downstream classifier is linear/MLP? This needs a discussion.
>
> A3. Most of the experiments in the paper are performed with linear classification protocol. The results have shown that under EqCo the number of negative samples does not affect much of the performance. The review may concern whether EqCo over fits the downstream linear classifier. We are pleased to provide the experiments with MLP instead of linear downstream classifier afterwards, but we think it is not likely.

---

### Official Review · AnonReviewer4 · 2020-11-06
**Good observation and experiments, but a bit over-claim**

**Rating:** 5
**Confidence:** 4

**Review:**

This paper introduces a margin term for positive pair in contrastive loss, and discovered an equivalent rule between margin m, temperature \tau, and number of negatives K. The authors demonstrate that once they set the three values following the equivalent rule, then relatively smaller number of negatives can also work well for contrastive learning.

Pros:
- It is interesting to introduce a margin m, and properly set a relation between it and temperature and number of negatives.
- Theoretically, the authors show that once you set the term contains K in the lower bound of MI to be constant, then the lower bound will be irrelevant to K. The equivalent rule is very interesting, but the theorem and Figure 1 is a bit trivial to me(see cons).
- The experimental results with MoCo, MoCo v2, and SimCLR shows that the equivalent rule improves the stability over different number of negatives for contrastive learning.

Cons:
- The reason why the lower bound now doesn't depends on K, is because you set the term that contains K as constant now. Now K is not an variable for the lower bound anymore, and the only variable in the lower bound is the density ratio, which is almost stable once the training converges. Also, theoretically the converged density ratio should be invariant to K. Therefore, you don't even need to plot Figure 1(b), but you will guess it's constant. To extreme, you can set I >= 0 = f_bound, but it tells you nothing. Thus,
(1) The real question is that, how this bound can sensitively and faithfully reflect the change of ground truth mutual information? To this end, I would recommend authors to follow the experiments in [A1] to generate variables that share different amount of mutual information, and check how your bounds track that. E.g., Figure 2 in A1.
(2) you can do such bound estimation on held-out data to check.
- I found myself not convinced by some statements in this paper:
(1) In abstract, "for the first time, we can perform self-supervised contrastive training using only a few negative pairs", some prior work [A2] has already reduce the number of negatives. So the tone here is a bit arrogant to me. Besides, BYOL paper has already shown that you can completely remove the denominator term in contrastive loss, and only do alignment.
(2) The first bullet of contribution section that says "large size of negative samples is critical" is a wrong interpretation. I kind of agree that small negative also works well, at the ImageNet scale. But what will happen for uncurated dataset whose size is far beyond ImageNet, is still unknown. So the claim here should be more restricted.
- The SIMO design choices you proposed here is exactly the same as SimCLR v2 [A3].
(1) No memory bank, key and negative from the same batch of momentum encoder. Yes, check SimCLR v2.
(2) SyncBN. Yes, it's in SimCLR and SimCLR v2.
- Missing related work:
(1) Big Self-Supervised Models are Strong Semi-Supervised Learners. NeurIPS 2020.
(2) Unsupervised Learning of Visual Features by Contrasting Cluster Assignments. NeurIPS 2020.
(3) What makes for good views for contrastive learning. NeurIPS. NeurIPS 2020.
(4) Whitening for Self-Supervised Representation Learning. arXiv

[A1] On Variational Bounds of Mutual Information. ICML 2019
[A2] Whitening for Self-Supervised Representation Learning.
[A3] Big Self-Supervised Models are Strong Semi-Supervised Learners. NeurIPS 2020.

Overall, I like the equivalent rule for stabilizing the training across different number of negatives. Meanwhile, I hope the authors could reconsider some statements. Also, it would be of great interest to see the experiments on estimating mutual information. I would consider changing my rating if my concerns get solved properly.


==== update ====

I found the bound the author provides is problematic (see my response for details). In short, the bound is problematic in that the bound will be more accurate when only one sample from the product of marginal (or negative) is used, i.e., $K=1$, while more negatives leads to less accurate bound. This definitely counters the intuition in theory where more samples should let you estimate the KL divergence between $p(x,y)$ and $p(x)p(y)$ better.

I believe there is a mathematical issue of directly setting up $m=\tau \log \frac{\alpha}{K}$ as in Eq(5). Though the empirical results look good, I recommend **NOT** accept papers with theory issues.

---

> ### Author Response · Authors · 2020-11-24
> **Response to AnonReviewer4**
>
> Thanks for the review comments.
>
> Q1. About the overclaiming issue.
>
> A1. We have revised those statements. For example, we remove the claim that "for the first time, we can perform self-supervised contrastive training using only a few negative pairs" in the abstract. We summarize our first contribution as "we interpret it from a
> different view: it may be because the hyper-parameters are not set to the optimum".
>
> Q2. The theorem and Figure 1 is a bit trivial to me. You don't even need to plot Figure 1(b), but you will guess it's constant. To extreme, you can set $\text{I} >= 0 = \text{f}_\text{bound}$, but it tells you nothing.
> The real question is that, how this bound can sensitively and faithfully reflect the change of ground truth mutual information.
>
> A2. First, we do not agree that the bound and Fig.1 is as trivial as $\text{I} >= 0 = \text{f}_\text{bound}$ .
> Notice that
>
> $ \text{log} (1 + K e^{m/\tau}) - \mathcal{L}_{NCE} \triangleq \hat{\text{f}}_\text{bound}$
>
> while $\mathcal{L}_{NCE}$ is still a function of $K$ and $m$ and needs to be optimized. When $m$ and $K$ changes, the landscape of the loss thus changes, we think it is still necessary to check whether $\hat{\text{f}}_\text{bound}$ converges to the similar values as expected, as shown in Fig.1. It is very different from the (trivial) concept of $\text{I} >= 0$ which even does not contain terms to be optimized.
>
> Second, the reviewer may concern about the tightness of the bound. It is known that the conventional InfoNCE estimator is of low variance but high bias. Our proposed EqCo just aims to simulate the behavior of InfoNCE loss using smaller $K$s, which does not help to reduce the bias. In Appendix C we introduce a verification following the settings of [A1] as suggested by the reviewer. Table 9 shows that with EqCo the estimation of MI keeps steady with different $K$s, which is able to simulate the conventional InfoNCE estimator under $K=512$ as we expected. In addition, both estimators reflect the change of ground truth MI. But, we also notice that when the ground truth MI is large (MI=10), the gap between the estimator and the ground truth also increases.
>
>
> Q3. The SIMO design choices you proposed here is exactly the same as SimCLR v2 [A3]. (1) No memory bank, key and negative from the same batch of momentum encoder. Yes, check SimCLR v2. (2) SyncBN. Yes, it's in SimCLR and SimCLR v2.
>
> A3. We add the discussion in Sec.3 and Footnote 6. We argue that the key difference is, the best configuration suggested by SimCLR v2 still uses memory bank. Though their experiments compare the one without memory bank, the score is worse. While in SiMo, thanks to EqCo we can remove the memory bank without accuracy drop. Despite all this, we have revised the statements to avoid misleading.
>
> Q4. Missing related work.
>
> A4. We have added the references. However, we have to point out [A2] is also a submission in ICLR 2021.

---

### Author Response · Authors · 2020-11-24
**Paper Revision**

Dear reviewers:

We have uploaded a new revision of our paper. Compared with the initial version, the change mainly includes:

A. Presentation

1) We revised some statements which may be misleading or overclaiming.

2) We add some descriptions and footnotes to justify the relations or differences to existing methods.

3) Some related works are added as suggested by R4.

B. Experiment

1) We add a new experiment on SimCLR, adjusting large $\alpha=4096$ to simulate large batch training while using small physical batch size $K=256$. Please refer to Fig.2(b) and the related text for details.

2) We add a new experiment to evaluate the InfoNCE lower bound with/without EqCo under different $K$s. See Appendix C for details.

---

### Decision · Program_Chairs · 2021-01-07
**Final Decision**

**Decision:**

Reject

**Comment:**

The equivalence rules for the margin are quite interesting, but I have two main concerns with the current paper (1) the theory does not seem to justify why increasing the number of negative examples helps in contrastive learning -- in fact Table 9 shows that the bound gets smaller as K increases. (2) The experiments use a large batch size (N) but use a smaller number of negative examples (K), which does not reduce the computation cost by much. The theoretical issue (1) can be a matter of improving the writing to put less emphasis on the theory. I invite the authors to address these issues and resubmit to other ML venues.

Detailed feedback: I believe you are missing a negative sign in your definition of optimal "loss" $\mathcal{L}_{opt}$ in Eq. (3), which resulted in AnonReviewer4's final comment.